# Optimizing Sugarcane Growth, Yield, and Quality in Different Ecological Zones and Irrigation Sources Amidst Environmental Stressors

**DOI:** 10.3390/plants12203526

**Published:** 2023-10-11

**Authors:** Muhammad Manzoor, Muhammad Zameer Khan, Sagheer Ahmad, Mashael Daghash Alqahtani, Muhammad Shabaan, Sair Sarwar, Muhammad Asad Hameed, Usman Zulfiqar, Sadam Hussain, Muhammad Fraz Ali, Muhammad Ahmad, Fasih Ullah Haider

**Affiliations:** 1Land Resources Research Institute, National Agricultural Research Centre, Islamabad 44000, Pakistan; manzoorm77@gmail.com (M.M.); zameerahmar@gmail.com (M.Z.K.); mshabaan@parc.gov.pk (M.S.); sardarsair@hotmail.com (S.S.); malik.asadhameed@gmail.com (M.A.H.); 2Pakistan Agricultural Research Council, Islamabad 45500, Pakistan; sagheersc@hotmail.com; 3Department of Biology, College of Science, Princess Nourah bint Abdulrahman University, P.O. Box 84428, Riyadh 11671, Saudi Arabia; 4Department of Agronomy, Faculty of Agriculture and Environment, The Islamia University of Bahawalpur, Bahawalpur 63100, Pakistan; 5College of Agronomy, Northwest A&F University, Xianyang 712100, China; ch.sadam423@gmail.com (S.H.); frazali15@gmail.com (M.F.A.); 6Department of Agronomy, University of Agriculture Faisalabad, Faisalabad 38040, Pakistan; ahmadbajwa516@gmail.com; 7Key Laboratory of Vegetation Restoration and Management of Degraded Ecosystems, South China Botanical Garden, Chinese Academy of Sciences, Guangzhou 510650, China; haider281@scbg.ac.cn; 8University of Chinese Academy of Sciences, Beijing 100039, China

**Keywords:** sugarcane, water quality, balanced fertilizer use, climatic conditions, environmental stressors

## Abstract

The imbalanced use of fertilizers and irrigation water, particularly supplied from groundwater, has adversely affected crop yield and harvest quality in sugarcane (*Saccharum officinarum* L.). In this experiment, we evaluated the impact of potassium (K) and micronutrients [viz. Zinc (Zn), Iron (Fe), and Boron (B)] application and irrigation water from two sources, viz. canal, and tube well water on sugarcane growth, yield, and cane quality under field trails. Water samples from Mardan (canal water) and Rahim Yar Khan (tube well water) were analyzed for chemical and nutritional attributes. The results revealed that tube well water’s electrical conductivity (EC) was three-fold that of canal water. Based on the EC and total dissolved salts (TDS), 83.33% of the samples were suitable for irrigation, while the sodium adsorption ratio (SAR) indicated only a 4.76% fit and a 35.71% marginal fit compared with canal water. Furthermore, the application of K along with B, Fe, and Zn had led to a significant increase in cane height (12.8%, 9.8%, and 10.6%), cane girth (15.8%, 15.6%, and 11.6%), cane yield (13.7%, 12.3%, and 11.5%), brix contents (14%, 12.2%, and 13%), polarity (15.4%, 1.4%, and 14%), and sugar recovery (7.3%, 5.9%, and 6%) in the tube well irrigation system. For the canal water system, B, Fe, and Zn increased cane height by 15.3%, 13.42%, and 11.6%, cane girth by 13.9%, 9.9%, and 6.5%, cane yield by 42.9%, 43.5%, and 42%, brix content by 10.9%, 7.7%, and 8%, polarity by 33.4%, 28%, and 30%, and sugar recovery by 4.0%, 3.9%, and 2.0%, respectively, compared with sole NPK application. In conclusion, the utilization of tube well water in combination with canal water has shown better results in terms of yield and quality compared with the sole application of canal water. In addition, the combined application of K and B significantly improved sugarcane yields compared with Zn and Fe, even with marginally suitable irrigation water.

## 1. Introduction

Water scarcity and quality have become serious issues for normal crop growth and sustainable agricultural development, which must be addressed through alternate water sources while ensuring soil fertility [1,2]. The long-term and excessive use of tube well water for irrigation, along with low precipitation and high evaporation, have increased salinity problems, resulting in low soil fertility and crop growth [2,3]. Applying highly saline irrigation water negatively impacts the soil-water-plant relationship, thereby restricting plants’ normal physiological functions [4,5]. Furthermore, it causes osmotic effects, water scarcity, nutritional imbalances, and oxidative stress, which also hinder sugarcane growth, leaf surface expansion, and metabolic activities [6]. Collectively, low water quality is the major limiting factor for sugarcane crop production [7,8]. In Pakistan, a severe imbalance exists between nutrients application through fertilization and their use efficiency [9], resulting in stagnant crop yield and a deteriorated quality of harvested produce. Moreover, the generally adopted practice of extracting nutrients from soil through continuous cropping system has progressively degraded soil quality [10].

Sugarcane (*Saccharum officinarum* L.) is a major crop cultivated worldwide owing to its enormous dietary and commercial applications. It is one of the most imperative industrial crops based on its extensive production in multiple tropical and sub-tropical regions on the globe [11]. The sugarcane industry contributes nearly 80% of total sugar [12]. Hence, about 28.3 million hectares in 90 countries are cultivated with sugarcane, which produces about 1.69 billion tons globally [13]. Sugarcane requires tropical and subtropical climates for normal growth and production [14]. Different factors perform pivotal roles in ensuring optimum yield and sugar production, these including climatic conditions [15], varieties [16], agronomic management practices [17], and soil fertility status [5]. Metrological factors such as temperature, rainfall, humidity, and sunshine significantly impacted sugarcane yield and sugar recovery. It is found that high temperature favors excellent growth and yield potential, whereas a semi-humid climate produces better sucrose contents [18]. 

Potassium offers a significant role in cane production as it enhances the crop’s resilience to intermittent drought conditions which commonly observed in sugarcane cultivated regions [19]. It also plays a pivotal role in controlling the stomatal aperture, thereby upholding turgor pressure during unfavorable moisture conditions. Under salinity stress, K application helps in preserving ion homeostasis and overseeing osmotic equilibrium [20]. Application of K has been reported to increase plant growth (by 19%), and count of millable cane attributable to the robust tillers developed in ratoon cane. The provision of sufficient K mitigates moisture stress [21] and salt stress in sugarcane [22]. It has been documented that distinct carrier proteins facilitate the transport processes, ensuring the crucial Na^+^/K^+^ equilibrium within cells, which is essential for plant survival in saline environments. Within plant cells, the vacuole serves as a reservoir of K^+^, performing a vital role in upholding cellular turgor [23].

The impact of K and micronutrient fertilization, particularly Fe, Zn, manganese (Mn), and B, on sugarcane is very significant and area-specific [24]. Incorporating micronutrients into sugarcane cultivation enhances the yield and plays a vital role in elevating the quality of juice. Applying Zn and Fe minerals also becomes imperative for achieving superior cane production over the long term and superior juice characteristics [25,26]. Similarly, sugar transport and cell wall development heavily rely on the presence of B in the plant’s system [27]. In this study, we hypothesized that applying K and different micronutrients with canal water irrigation could improve sugarcane growth, yield, and quality attributes. Despite a known fact regarding mixing tube well and canal waters for managing salinity, limited literature evidence exists about the combined application of K and different micronutrients under various irrigation schemes. In the current study, we evaluated the combined efficacy of K and other micronutrients in improving sugarcane growth, yield, and quality. Considering the importance of irrigation water with K and micronutrients on sugarcane growth, yield, and quality, the current study was conducted to investigate irrigation water characteristics, i.e., canal water from the Sawat River and tube well water from Rahim Yar Khan with varying climatic conditions.

## 2. Results

### 2.1. Characteristics of Irrigation Water and Nutrient Status

Evaluation of irrigation water from two sources showed a diverse variation regarding chemical characteristics and nutrient supply to crops. Results regarding water characteristics revealed that the EC of tube well water ranged from 0.218 to 5.34 dSm^−1^, with an average of 1.09 dSm^−1^, while canal water’s EC varied between 0.08 to 1.08 dSm^−1^, presenting an average of 0.24 dSm^−1^ showing high variability, i.e., 84% and 107%, respectively (Table 1). The EC of tube well water was almost three times higher than canal water. Based on EC, 83.33% of the total samples collected from tube well (Rahim Yar Khan) were suitable for sugarcane irrigation purposes, whereas 14.29% were marginally fit and 2.38% were unfit. In comparison, 100% of water samples from canal water (Mardan and Charsadda) were suitable for sugarcane irrigation. In the case of soluble salts, Na in water ranged from 27 to 398 mg L^−1^ for tube well sources compared with canal water (1.47 to 19.5 mg L^−1^). Similarly, Ca^2+^, Mg and CO_3_ + HCO_3_ contents varied between 74.56–885.4 and 61–406.9 mg L^−1^ in tube well water compared with canal water (42.6–160 and 40.9–447) with the mean values of 190.74–187.34 mg L^−1^ and 94.32–131.4 mg L^−1^ in tube well and canal water, respectively. Chloride contents in both water sources depicted that tube well water contained a high concentration (60–720 mg L^−1^) compared with canal water (0.67–180 mg L^−1^). Analysis of these water samples revealed that N, P, and K were 0.53, 0.88, and 25.05 mg L^−1^ in tube well water, whereas 1.36, 0.71, and 56.85 mg L^−1^ in canal water, respectively, indicating the suitability of canal water for sugarcane cultivation when compared with tube well water concerning soil health, crop growth, yield, and quality.

### 2.2. Growth and Yield of Sugarcane

It was also reported that applying soil K either alone or in combination with micronutrients, such as Zn, Fe, and B, led to a statistical significant (*p* ≤ 0.05) inrement in sugarcane height, girth, and yield compared with conventional farming practices (Table 2). Applying K significantly increased cane height (8.4% and 9.74%) and girth (9.74% and 9.39%). Soil application of K and foliar application of B further increased cane height by 12.81% and 13.42% at Rahim Yar Khan and Mardan sites over conventional farming practices. Similarly, upon integrating K with micronutrients, both sites increased cane girth by 13.42% and 14.32%, respectively. Similarly, K application increased yield by 10.38% and 40.79% over conventional practices for Rahim Yar Khan and Mardan sites, respectively. The combined application of micronutrients and K increased cane yield by 12.49% and 42.79% compared with conventional farming practices. Soil applied K with foliar application of B showed maximum (197,767 and 198,826) millable canes (Figure 1) compared with sole NP (183,824 and 187,390) with resultant increases of 7.6% and 5.7% at both sites, respectively. The number of internodes in response to K application were improved by 12.2% and 9.3%, whereas the B application increased by 15.2% and 15.4% over conventional farming practices at Rahim Yar Khan and Mardan sites, respectively.

### 2.3. Quality Characteristics

Quality attributes of sugarcane in response to different sources of irrigation water along with K and micronutrient application showed that brix contents, polarity, and sugar recovery were significantly (*p* ≤ 0.05) improved (Table 3). Brix contents in conventional farming practices were 15.75% and 19.24% for both sites. Soil application of K with Zn increased brix content by 20.78% with the canal water irrigation at the Mardan site compared with 17.67% at the Rahim Yar Khan Site irrigated with tube well and canal water in combination (Figure 2). In addition, K and B application increased brix content by 12.4% over conventional farming practices. Similarly, adding K with B at the Mardan site yielded the highest (18.67%) polarity by canal water irrigation. In sugar recovery, conventional farming practices resulted in the lowest (10.28%), which increased by 13.4% after adding K with Fe and B (Figure 1). Site effects revealed that sugar recovery was 17.5% higher at the Mardan site irrigated with canal water than Rahim Yar Khan Site irrigated with canal water and tube well water combined. For sugar production, K application in combination with B increased yield by 22% and 49%, respectively, at the Rahim Yar Khan and Mardan sites, with relative increases of 22% and 49% over conventional farming practices (Table 2). The response to K and micronutrients was inverted in the case of fiber contents where soil treated with NP as conventional farming practices showed the highest fiber contents, i.e., 12.29% and 12.19%, which reduced to 12.03% and 11.67% in soil applied K with Fe at Rahim Yar Khan and Mardan sites, respectively (Figure 3). Thus, results revealed that using K and micronutrients reduced the fiber contents and increased sugar production. Similarly, canal water decreased fiber contents and improved sugar production compared with tube well and canal water combined. It was found that chlorophyll contents were increased by 31.4 to 42.5% and 33.8 to 43.8% at both sites, respectively, in response to combined application of K and Fe, over conventional farming practices (Figure 3). Maximum juice purity (84.9%) was recorded for Rahim Yar Khan site followed by Mardan site with soil-applied K along with foliar Zn application and sole K application, respectively, where it was recorded that canal water irrigation was better in term of juice purity and sugar production than canal and tube well water (Figure 3).

### 2.4. Potash and Micronutrients

Soil applied K and micronutrients showed a significant impact on K contents at both sites, resulting in a significant increase in K content in leaf blade (15.3%) and leaf sheath of sugarcane (2.7%) over conventional farming practices (Figure 4). The values were further increased by 16.3% and 4.0% after adding Fe to K in both plant tissues at the Rahim Yar Khan site. For Mardan site, the application of K alone showed a maximum increase of 41.6% and 13.5% in leaf blade and leaf sheath, respectively. Sugarcane micronutrient contents such as Zn, Fe, and B in response to soil K application were remarkably improved by 26.3%, 59%, and 14.5% at the Rahim Yar Khan site, whereas at the Mardan site, values were increased by 123%, 104%, and 16.5% in leaf blade as well as leaf sheath, respectively, as compared with conventional farming practices (Figure 4). The combined application of K with Zn resulted in 34% and 37% increase at Rahim Yar Khan, while at the Mardan site, 144% and 160% in leaf blade and leaf sheath Zn contents, respectively. Similarly, Fe contents were increased by 70% and 50% at Rahim Yar Khan and 21% and 22.5% at the Mardan site in both plant parts. The combined application of K and B resulted in 247% and 152% increase in B content (Rahim Yar Khan) and 86% and 167% (Mardan) in leaf blade and leaf sheath of sugarcane, respectively, compared with conventional farming practices. Thus, it was found that soil application of sole K effectively improved K uptake, which, combined with Zn, Fe, and B, further improved the respective micronutrient uptake.

## 3. Discussion

Irrigation water is crucial in influencing soil health and crop quality and is categorized into suitable, marginal, and unsuitable classes based on its EC and TDS [27]. Our study showed that EC and TDS levels of tube well water were nearly three times higher than canal water. Consequently, tube well water was deemed unsuitable for irrigation purposes. These findings are similar with Feng et al. [28]. Furthermore, previous research has also indicated that using saline water for irrigation can lead to soil salinity and increased pH [27]. Similarly, our investigations revealed that the SAR of water samples from tube wells was substantially higher than canal water, rendering it unsuitable for irrigation purposes. Similar results were previously reported by Kumar et al. [27] who stated that water samples from tube wells water source exhibited substantially higher SAR. Irrigation with saline water that exceeds threshold levels for EC, SAR, and TDS has detrimental effects on plant growth and development, resulting in reduced crop yields. Previous studies have demonstrated that K is vital in supporting plants during stress by maintaining ion equilibrium and regulating osmotic balance [29,30,31,32]. Our study further recorded that K and micronutrient applications improved sugarcane growth and yield under canal water irrigation schemes compared with conventional farming practices. 

Extreme environmental events associated with climate change are becoming more frequent worldwide and could push roughly one-third of global food crop and livestock production beyond the safe climatic space by the end of this century [33,34,35,36,37,38,39,40,41,42,43,44]. Salinity stress harms photosynthesis and the translocation of photoassimilates in sugarcane. Pabuayon et al. [45] stated that salinity negatively affects growth and development. However, K plays a crucial role in regulating stomatal aperture, thereby maintaining turgor pressure even under less favorable soil moisture conditions. It was worth noticing that even lower cane yield and higher sugar contents, as observed at the Mardan site, produced sugar at par with the higher cane yield at the Rahim Yar Khan site. Similar reports have highlighted that sustainable sugar production with improved sucrose content is achieved through suitable irrigation water and appropriate fertilizer management [46]. As reported earlier, changes in temperature and rainfall during critical stages resulted in a significant variation in crop yield and sugar recovery in three provinces of Pakistan, ultimately affecting overall production [47]. Jaiswal et al. [48] described that K application to sugarcane resulted in a 19% increase in overall plant growth compared with soil without K fertilization. Thus, adequate K supply to sugarcane reduces moisture stress, induces salt tolerance to sugarcane [16,21], and produces higher stalk and cane yields [47,48]. 

Furthermore, K application reduces N requirements through better N metabolism and transportation of nitrate to shoots [49,50]. A foliar spray of micronutrients combined with soil-applied K also improved cane yield (12.49% and 42.79% over conventional farming practices) but also yielded the higher brix contents and sugar recovery [51,52]. Similar was also reported for other crops [53]. Our results were supported by Ismail et al. [54], that Zn and Fe applied as foliar to the sugarcane effectively improved the growth characteristics of sugarcane, which ultimately increased its yield. Ghaffar et al. [25] described that sugarcane growth and yield traits were significantly improved due to applied Zn and Fe. Zhao et al. [55] reported that K application improved sugarcane growth under saline conditions. Franco et al. [56] stated that Zn fertilization enhanced stalk weight and cane quality. The brix contents in soil-applied K and B in combination with canal water showed high brix contents and sugar recovery. Although Watanabe et al. [57] correlated elevated K levels with a decrease in sucrose content. Zhao et al. [55] supported our results that K supplied either alone or in combination with Zn resulted in improved brix and sugar recovery. Potash with micronutrients is a balanced and complete package of sugarcane nutrition as, according to our findings. Thus, it was found that the soil application of K alone effectively improved K uptake, which, combined with Zn, Fe, and B, further improved the respective micronutrient uptake. The better uptake of micronutrients contributed to the heightened activity of sucrose-synthesizing enzymes, which also improved yield and sucrose content [57]. Iron is an essential micronutrient exhibiting a synergistic interaction with K [58,59]. Zinc is a critical component of numerous enzymes and proteins. As reported in the previous findings, K, Zn, and B uptake in sugarcane leaf tissues has effectively been improved by soil applied K and micronutrient integration [59]. 

## 4. Materials and Methods

A comprehensive study was conducted to survey sugarcane-growing districts such as Rahim Yar Khan in Punjab province and Mardan/Charsadda in Khyber Pakhtunkhwa (KPK) province. These districts are significant sugarcane-growing areas in their respective provinces and are known for utilizing different water sources for sugarcane irrigation. Subsequently, two specific fields were identified within these selected districts for conducting field experiments. These fields were situated on farmer-owned land and received irrigation water from distinct sources. In Mardan, the irrigation source was the Swat River canal, while in Rahim Yar Khan, the fields received a combination of canal and tube well water for irrigation, with alternating irrigation timings. 

### 4.1. Collection of Water Samples and Characteristics Studied

Forty-two union councils were selected from District Rahim Yar Khan, and 52 were chosen from Mardan and Charsadda districts. To ensure the integrity of the water samples, plastic bottles were used, which were thoroughly rinsed five times with tap water, followed by distilled water. For tube well water, samples were collected after allowing the well to run for 15 min to stabilize. These collected samples were then transported to the Soil Fertility Laboratory of the Soil-Plant Nutrient Program at NARC, Islamabad. Here, the samples underwent processing and analysis for various parameters, including pH, electrical conductivity (EC), and levels of N, P, K, Ca^2+^, Mg, CO_3_, HCO_3_, Na, B, and Cl^−^, by following standard protocols [49]. Data obtained from these analyses were utilized to assess the nutrient status and determine the extent of different nutrient-related disorders.

### 4.2. Experimental Sites and Treatments

Field experiments were conducted during the 2021–2022 seasons at two selected sites on farmer fields where sugarcane was cultivated. In February 2021, sugarcane plantation was prepared before pre-plantation soil sampling was conducted to determine soil physicochemical traits and fertility status (Table 4). The treatment combinations were: T1 = local farmer practice; N and P application (200 and 100 kg ha^−1^), T2 = N + P + K (200 + 100 + 150 kg ha^−1^), T3 = N + P + K (200 + 100 + 150 kg ha^−1^) + Zn (0.1%) − 3 Foliar Spray, T4 = N + P + K (200 + 100 + 150 kg ha^−1^) + Fe (0.1%) − 3 Foliar Spray and T5 = N + P + K (200 + 100 + 150 kg ha^−1^) + B (0.05%) − 3 Foliar Spray. During the seeding process, fertilizers P and K were applied, while N was administered in three separate doses: during seeding, tillering, and knot formation stages. Micronutrients (Fe, Zn, and B) were applied through three foliar sprays, beginning at a height of 92 cm and continuing at one-month intervals after the first application. The experimental units were arranged using a randomized complete block design (RCBD) comprising five treatments, each with three replications. The plot size was maintained at 70 × 10 m for each treatment and replication. Planting was conducted at a spacing of 0.726 m (R × R) distance with four budded sets placed in a single row. The sugarcane variety ‘CP-77400’, approved uniformly for both provinces (Punjab and Khyber-Pakhtunkhwa), was selected as a test cultivar due to its adaptability and uniformity in agronomic, physiological, yield, and quality traits. One month after the completion of fertilizer application, chlorophyll contents were measured using a chlorophyll meter (SPAD-502), and plant tissue samples were collected in August 2022 to assess nutrient status at the grand growth stage. Tissue sampling involved collecting ten samples from each replication of one treatment, taken from the middle three rows. The fifth and sixth leaves from the top of the cane were collected, and leaf sheaths and leaf blades were separated, washed, air-dried, and oven-dried at 65 °C for 24 h. Data regarding growth and yield components, including cane height, cane girth, number of internodes, millable canes, and cane quality traits such as brix contents, polarity, juice purity, fiber contents, and sugar recovery, were recorded at the end of December upon weighing the fresh millable canes harvested from three inner rows of each plot. Brix contents (%), polarity (%), juice purity (%), and fiber contents (%) were quantified in a sugar mill cane quality laboratory. Sugar recovery (%) was calculated from brix, purity, polarity, and fiber contents, whereas sugar production (T ha^−1^) was determined by multiplying cane yield by sugar recovery.

### 4.3. Plant Tissue Analysis and Quality Characteristics

Plant tissue samples collected at the grand growth stage underwent analysis for their N, P, K, Zn, Fe, and B contents. Samples obtained from the field, specifically leaf sheath and leaf blade, were subjected to oven drying, followed by grinding and sieving. The total N content was determined using the Kjeldahl method, while P, K, Zn, and Fe were analyzed in a di-acid digestion mixture with a ratio of 2:1 (HNO_3_:HClO_4_). Boron content was determined through dry ashing and subsequent measurement [60].

### 4.4. Statistical Analysis

Data obtained were statistically analyzed using STATISTIX 8.1 software. For water characteristics, we analyzed samples to determine the range, calculate average values, and assess variability. For field experiments, data were analyzed for analysis of variance (ANOVA) in RCBD design with two-factor factorial, and a Least Significant Difference (LSD) was calculated at a 5% significance level.

## 5. Conclusions

Water quality has a direct impact on soil properties and its suitability for sugarcane cultivation. Water quality, soil conditions, and soil fertility directly affect plant growth and yield response. The present study investigated the relationship between K and different micronutrients instead of improving sugarcane growth, yield, and quality under two irrigation schemes (canal and tube well water). We concluded that canal water contributes to superior sugarcane quality with increased yield and improved cane quality over tube well water. Furthermore, using K and micronutrients (Zn, Fe, and B) may support sugarcane in enduring and yielding more effectively even under irrigation water’s salt stress conditions. However, we need to expand our experimental area towards more heterogeneous agro-ecological areas to validate these results. Since sugarcane is one of the major crops worldwide, applying these treatments under diverse climate zones can help researchers enhance the sugarcane quality, yield, and even nutrient use efficiencies under those regions.

## Figures and Tables

**Figure 1 plants-12-03526-f001:**
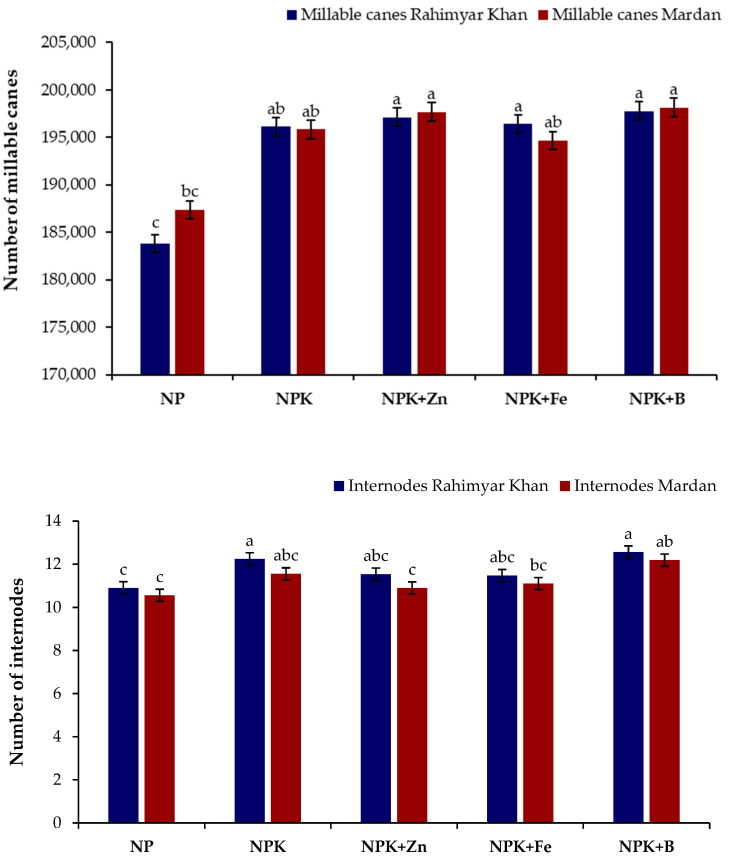
Effect of potash (K) and micronutrients on millable canes and number of internodes. Different lowercase letters indicate significant difference among nutrients application. N, nitrogen; P, phosphorus; K, potassium; Zn, zinc; Fe, iron; B, boron.

**Figure 2 plants-12-03526-f002:**
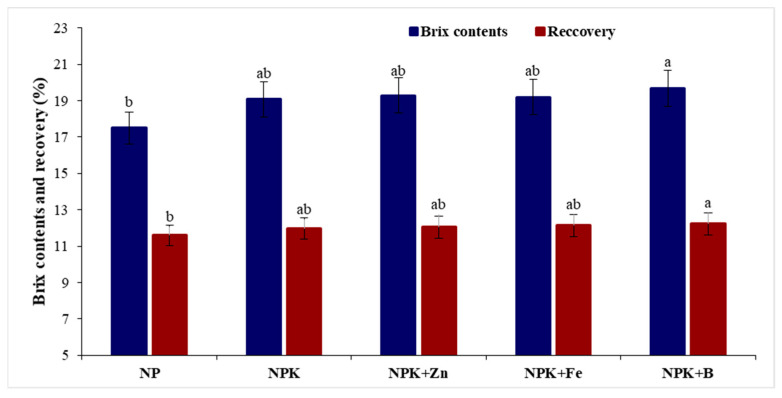
Effect of potash (K) and micronutrients on brix contents and sugar recovery average over two sites. Different lowercase letters indicate significant difference among nutrients application. N, nitrogen; P, phosphorus; K, potassium; Zn, zinc; Fe, iron; B, boron.

**Figure 3 plants-12-03526-f003:**
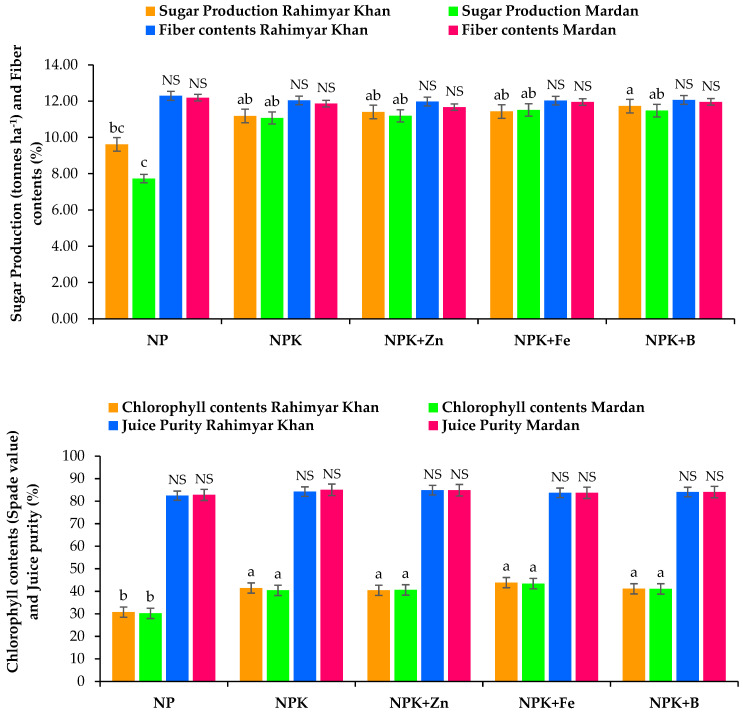
Effect of potash (K) and micronutrients on sugar production, fiber contents, chlorophyll contents, and juice purity of sugarcane at the Rahim Yar Khan and Mardan sites. Different lowercase letters indicate significant difference among nutrients application. N, nitrogen; P, phosphorus; K, potassium; Zn, zinc; Fe, iron; B, boron.

**Figure 4 plants-12-03526-f004:**
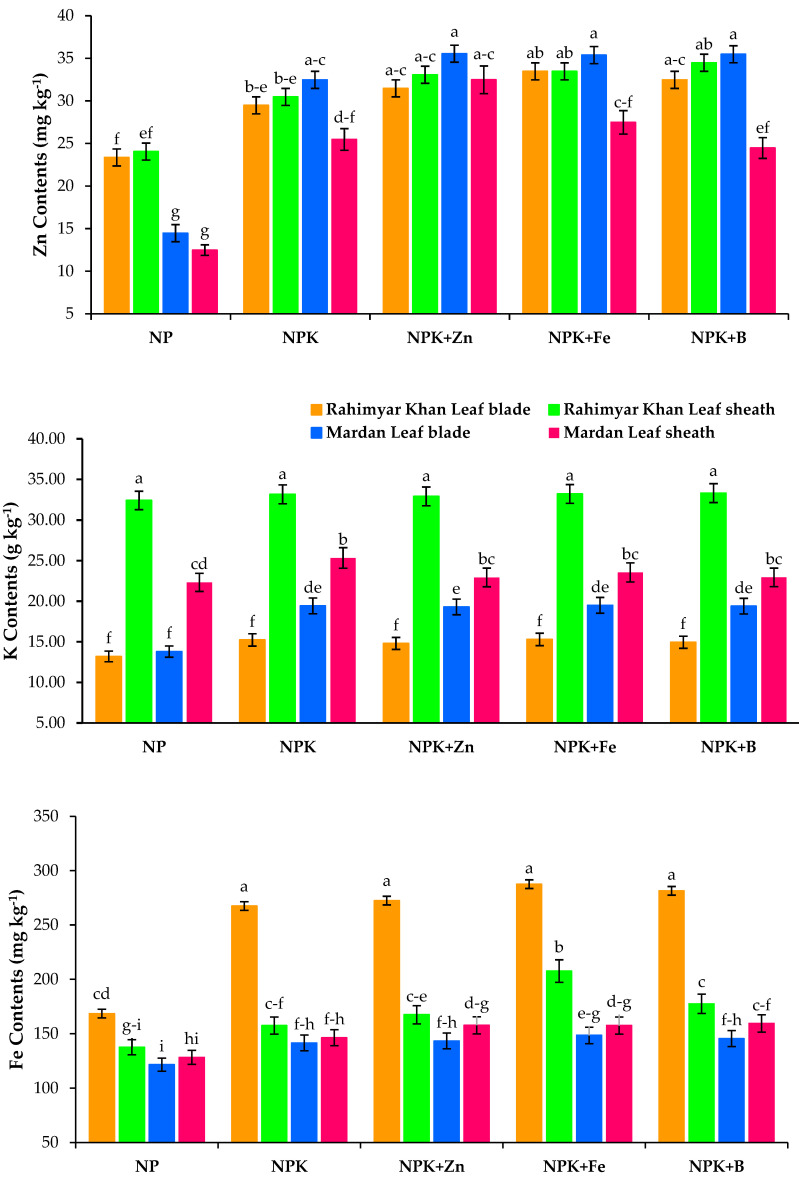
Effect of potash (K) and micronutrients on K, Zn, Fe, and B contents of sugarcane leaf blade and leaf sheath at Rahim Yar Khan and Mardan sites. Different lowercase letters indicate significant difference among nutrients application. N, nitrogen; P, phosphorus; K, potassium; Zn, zinc; Fe, iron; B, boron.

**Table 1 plants-12-03526-t001:** Chemical characteristics and nutrient status of tube well and canal water surveyed from Districts Rahim Yar Khan (Punjab) and Mardan/Charsadda (Khyber Pakhtunkhwa).

Characters	Mean		Range		SD		SEM		CV	
	Tubewell	Canal	Tubewell	Canal	Tubewell	Canal	Tubewell	Canal	Tubewell	Canal
EC	1.09	0.24	0.218–5.34	0.08–1.08	0.912	0.256	0.14	0.036	84.14	107.3
Na	174.5	10.49	27–398	1.47–19.5	112.01	4.83	17.29	0.669	64.20	46.03
Ca + Mg	190.74	94.32	74.56–885.4	42.6–160	160.9	34.42	24.83	4.77	84.37	36.49
CO_3_^+^ HCO_3_^−^	187.34	131.4	61–406.9	40.9–447	77.44	92.1	11.95	12.76	41.34	70.01
Cl	172.38	78.64	60–720	0.67–180	132.5	33.41	20.45	4.632	76.87	42.48
TDS	714.4	153.2	139.5–4272	48.6–693	686.9	164.4	106	22.79	96.16	107.3
NO_3_-N	0.532	1.36	0.39–0.94	0.33–3.39	0.148	0.893	0.023	0.123	27.9	65.51
P	0.881	0.71	0.58–2.79	0.37–1.26	0.361	0.167	0.056	0.023	40.9	23.8
K	25.05	56.85	13–46	29–87	7.37	13.8	1.14	1.91	29.4	24.3

Note: EC = Electrical conductivity, Na = Sodium, Ca + Mg = Calcium and magnesium, CO_3_ + HCO_3_^−^ = Carbonates and bicarbonates, Cl = Chlorides, TDS = Total Issolved Salts, NO_3_-N = Nitrate nitrogen, P = Phosphorus, K = Potassium

**Table 2 plants-12-03526-t002:** Growth and yield of Sugarcane in response to potash (K) and micronutrient application at two locations with different irrigation water sources.

	Cane Height (cm)			Cane Girth (cm)			Cane Yield (Tonnes ha^−1^)		
	RYK	Mardan	Mean	RYK	Mardan	Mean	RYK	Mardan	Mean
NP	299.15 ab	227.03 d	263.09 B	7.7 ab	6.77 d	7.235 B	93.54 ab	59.86 c	76.71 B
NPK	324.27 a	249.15 cd	286.71 A	8.43 a	7.37 cd	7.90 A	103.25 a	84.32 b	93.79 A
NPK + Zn	330.82 a	253.28 bc	292.05 A	8.60 a	7.21 bc	7.905 A	104.29 a	85.03 b	94.66 A
NPK + Fe	328.49 a	257.49 bc	292.99 A	8.91 a	7.44 bc	8.175 A	105.07 a	85.88 b	95.47 A
NPK + B	337.48 a	261.71 bc	299.595 A	8.93 a	7.71 bc	8.32 A	106.32 a	85.53 b	95.92 A
LSD (*p* ≤ 0.05)	48.25		28.78	1.89		1.13	14.21		8.48
Site effect	324 A	244 B	12.66	8.51	8.09	NS	102.49 A	80.12 B	3.73

Uppercase letters indicate grand means; lowercase letters indicate individual means. LSD, least significant difference at 5% level of significance; N, nitrogen; P, phosphorus; K, potassium; Zn, zinc; Fe, iron; B, boron.

**Table 3 plants-12-03526-t003:** Quality characteristics of sugarcane in response to potash (K) and micronutrients application at two locations with different irrigation water sources.

	Brix (%)			Polarity (%)			Sugar Recovery (%)		
	RYK	Mardan	Mean	RYK	Mardan	Mean	RYK	Mardan	Mean
NP	15.75 b	19.24 ab	17.50 B	12.86 b	14.00 ab	13.43	10.28 AB	12.91 B	11.60 B
NPK	17.49 ab	20.65 ab	19.07 AB	14.63 ab	18.39 ab	16.51	10.83 AB	13.13 A	11.98 AB
NPK + Zn	17.79 ab	20.78 ab	19.29 AB	14.68 ab	18.20 ab	16.44	10.94 AB	13.16 AB	12.05 AB
NPK + Fe	17.67 ab	20.72 ab	19.20 AB	14.66 ab	17.97 ab	16.31	10.88 AB	13.41 A	12.15 AB
NPK + B	17.99 ab	21.33 a	19.66 A	14.84 a	18.67 a	16.75	11.03 AB	13.42 A	12.23 A
LSD (*p* ≤ 0.05)	5.99		NS	5.61		NS	2.93		1.75
Site effect	17.34 B	20.75 A	1.57	14.33 B	17.44 A	1.47	10.85 B	12.75 A	0.77

Uppercase letters indicate grand means; lowercase letters indicate individual means. LSD, least significant difference at 5% level of significance; N, nitrogen; P, phosphorus; K, potassium; Zn, zinc; Fe, iron; B, boron.

**Table 4 plants-12-03526-t004:** Soil physico-chemical characteristics and nutrient status of selected sites.

Location	Depth	Texture	pH	EC	OM	NO_3_-N	P	K	Fe	Zn	B
Rahim Yar Khan	(inches)		1:1	dSm^−1^	(%)	(mg kg^−1^)					
	0–30	Silt loam	8.2	0.512	0.7	3.2	0.78	99	6.70	0.40	1.05
	30–45	-	8.5	0.410	0.4	4.6	0.77	102	5.03	0.60	1.30
Mardan	0–30	Silt	8.22	0.414	0.3	3.9	2.6	168	6.5	0.6	0.61
	30–45	-	8.18	0.885	0.20	1.3	2.4	198	3.4	0.7	0.6

## Data Availability

Not applicable.

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
