# Peer review of "Optimizing Sugarcane Growth, Yield, and Quality in Different Ecological Zones and Irrigation Sources Amidst Environmental Stressors"

_plants, 2023, doi:10.3390/plants12203526_

Round 1
Reviewer 1 Report
Overall, this research article represents an interesting investigation on “Optimizing Sugarcane Growth, Yield, and Quality in Different Ecological Zones and Irrigation Sources amidst Environmental Stressors.” The manuscript presents an intriguing study that uses a comprehensive approach by examining the impact of potassium (K) and micronutrient applications along with different irrigation sources on various aspects of sugarcane growth, yield, and quality attributes. It addresses a significant issue in agriculture by exploring strategies to optimize sugarcane cultivation in the face of environmental stressors and imbalanced fertilizer use, which is crucial for sustainable farming. However, a few essential points need to be addressed before the manuscript can be considered for publication. Below are some specific comments for the authors' consideration:
The manuscript provides a comprehensive introduction to the issues surrounding sugarcane irrigation and fertilizer use. However, it would be beneficial to include some global context or statistics on sugarcane production and its significance to set the stage for the study.
The authors conducted field experiments in two different locations with varying climates i.e., Mardan and Rahim Yar Khan. While the inclusion of multiple locations is a strength, there are critical aspects that need further attention:
· Climate Influence: The study acknowledges the use of different ecological zones but lacks a comprehensive analysis of the potential impact of climate variations on the study's outcomes. A more in-depth exploration of how specific climate factors may have influenced sugarcane growth and quality in each location is crucial. It is essential to discuss temperature, precipitation, and sunlight variations and their potential effects on the results.
· Varietal Selection: The manuscript does not provide information about the specific sugarcane varieties used in the experiments. Different varieties respond differently to environmental conditions, and it is vital to clarify the chosen varieties. Failure to account for varietal adaptability may introduce a confounding variable in the study.
The manuscript lacks crucial information regarding the size and design of the experimental plots used in the field experiments conducted at Mardan and Rahim Yar Khan. The absence of details related to plot size and layout raises questions about the experimental design's robustness and the reliability of the results.
The manuscript does not provide information on the sample size used in the field trials. It's crucial to ensure that the sample size is adequate to draw meaningful conclusions and that the samples are representative of the ecological zones under investigation.
The manuscript mentions that "Quality parameters were analyzed by sugar mills located in the pertinent regions engaged in sugarcane production." While it's acknowledged that sugar mills often play a role in quality assessment, the absence of specific details about the quality parameter analysis in the manuscript is a significant limitation. For the sake of clarity, transparency, and the ability of readers to fully comprehend the study, it is imperative that the authors provide more detailed information on the quality parameters examined, the specific methods and instruments used for analysis, and the criteria or standards against which the results were evaluated. Without this information, it is challenging to assess the validity of the quality parameter results and their significance in the context of sugarcane cultivation.
To strengthen the paper, consider discussing the limitations of the study, potential avenues for future research, and any practical recommendations for sugarcane growers.
While the findings are interesting, it's essential to emphasize the novelty or uniqueness of the study's contribution to the field of sugarcane cultivation. How does this research advance our understanding or offer practical solutions to existing challenges in sugarcane farming?
Overall, this manuscript addresses an important agricultural issue and presents valuable findings. With some major revisions and additional context, it could make a significant contribution to the field of sugarcane cultivation and sustainable agriculture.
Author Response
SPECIFIC COMMENTS
Overall, this research article represents an interesting investigation on “Optimizing Sugarcane Growth, Yield, and Quality in Different Ecological Zones and Irrigation Sources amidst Environmental Stressors.” The manuscript presents an intriguing study that uses a comprehensive approach by examining the impact of potassium (K) and micronutrient applications along with different irrigation sources on various aspects of sugarcane growth, yield, and quality attributes. It addresses a significant issue in agriculture by exploring strategies to optimize sugarcane cultivation in the face of environmental stressors and imbalanced fertilizer use, which is crucial for sustainable farming. However, a few essential points need to be addressed before the manuscript can be considered for publication. Below are some specific comments for the authors' consideration:
Response: Thank you for sparing your valuable time to review the current manuscript. Your comments and suggestions will surely improve the quality of the manuscript. Below is the response to your queries. Moreover, these comments have been incorporated and responded to in the original manuscript in track changes.
Comment: The manuscript provides a comprehensive introduction to the issues surrounding sugarcane irrigation and fertilizer use. However, it would be beneficial to include some global context or statistics on sugarcane production and its significance to set the stage for the study.
Response:
Thank you for your insightful feedback. We truly appreciate your valuable suggestions. In response to your comments, we have taken the opportunity to enhance the introductory section of the manuscript. We believe these revisions will provide a more comprehensive context for readers by shedding light on the global significance of sugarcane production. In the revised version of the manuscript, we have incorporated the latest data on sugarcane production and its importance, as suggested. Here is the improved passage:
"Sugarcane is a pivotal crop on the global agricultural landscape due to its immense significance in dietary consumption and commercial applications. Its prominence extends to being one of the foremost industrial crops, primarily due to its extensive cultivation in numerous tropical and sub-tropical regions worldwide. Remarkably, the sugarcane industry contributes nearly 80% of the total sugar production on a global scale. This agricultural powerhouse covers approximately 28.3 million hectares of land across 90 countries, culminating in a staggering total production of 1.69 billion tons of sugarcane annually."
We hope these revisions successfully address your concern and enrich the manuscript by providing a more comprehensive global context for the study. Once again, we appreciate your valuable input, which has undoubtedly improved the overall quality of our work.
Comment: The authors conducted field experiments in two different locations with varying climates i.e., Mardan and Rahim Yar Khan. While the inclusion of multiple locations is a strength, there are critical aspects that need further attention.
Response:
Thank you for your valuable feedback. We appreciate your recognition of the strength of including multiple locations with varying agro-ecological climates in our study. Incorporating Mardan and Rahim Yar Khan allowed us to assess the impact of our applied treatments under different climatic conditions, enhancing our findings' robustness.
Regarding your concern, we agree that critical aspects require further attention. In our current research, we focused on comparing tube wells and canal water as individual sources of irrigation, as well as their combined use, to examine their effects on various attributes of sugarcane. While we obtained interesting results from these studies, we acknowledge that certain critical aspects still need to be addressed.
We would like to inform you that we are working on a follow-up manuscript addressing these critical aspects comprehensively. This manuscript is currently in preparation, and we are committed to conducting additional experiments and analyses to provide a more comprehensive understanding of the subject matter. We will ensure that the forthcoming manuscript addresses your highlighted limitations and incorporates the necessary data and discussions to provide a more holistic perspective on our research.
.
Comment: Climate Influence: The study acknowledges the use of different ecological zones but lacks a comprehensive analysis of the potential impact of climate variations on the study's outcomes. A more in-depth exploration of how specific climate factors may have influenced sugarcane growth and quality in each location is crucial. It is essential to discuss temperature, precipitation, and sunlight variations and their potential effects on the results.
Response: We appreciate your thoughtful comments on our study and would like to address your concerns regarding the potential impact of climate variations on our research outcomes. Our primary research focus in this study was to investigate the role of potassium and micronutrients in mitigating stress and improving sugarcane production. Additionally, we aimed to compare the suitability of canal water versus tube water for optimal sugarcane practices. Mardan and Rahim Yar Khan are prominent sugarcane production regions in Pakistan, each with unique challenges and opportunities.
We acknowledge that climatic conditions vary significantly from region to region, and we agree that a comprehensive analysis of climate factors could provide valuable insights. However, our study was designed with a specific research agenda in mind, and we chose to avoid incorporating climatic data as we wanted to maintain the focus on our core research themes.
We believe that future research endeavors could delve into the influence of specific climate factors, such as temperature, precipitation, and sunlight variations, on sugarcane growth and quality in these regions. Such an analysis would complement our study and provide a more holistic understanding of the factors affecting sugarcane production.
Comment: Varietal Selection: The manuscript does not provide information about the specific sugarcane varieties used in the experiments. Different varieties respond differently to environmental conditions, and it is vital to clarify the chosen varieties. Failure to account for varietal adaptability may introduce a confounding variable in the study.
Response: We sincerely thank the reviewer for their insightful comments regarding the inclusion of the specific sugarcane variety used in our manuscript. We recognize the importance of this information in understanding the potential variations in response to environmental conditions, and we have taken steps to address this concern.
In our study, we exclusively utilized the sugarcane variety "CP-77400," which holds uniform approval for cultivation in both provinces, Punjab and Khyber-Pakhtunkhwa. To ensure clarity and transparency, we have incorporated the name of this variety in the revised manuscript. Specifically, we have included this vital detail in the "Experimental sites and locations" section of the Materials and Methods, per the reviewer's valuable suggestion.
We believe that this addition significantly enhances the comprehensibility of our research, addressing the potential confounding variable of varietal adaptability. We sincerely appreciate the reviewer's input, which has improved our manuscript. If there are any further concerns or suggestions, please do not hesitate to let us know. Your feedback is invaluable to us, and we are committed to enhancing the quality and clarity of our work.
Comment: The manuscript lacks crucial information regarding the size and design of the experimental plots used in the field experiments conducted at Mardan and Rahim Yar Khan. The absence of details related to plot size and layout raises questions about the experimental design's robustness and the reliability of the results.
Response: Thank you for your feedback. In response, we have added essential details to the manuscript regarding the experimental plot size and design. We followed a Randomized Complete Block Design (RCBD) with five treatments, each having three replications. The plot size was consistently 70×10 meters for each treatment, with one replication per plot. Plantation was conducted with a spacing of 0.726 meters between rows and using a four-budded setts arrangement in a single-row placement. These details improve the clarity and reliability of our experimental design. We appreciate your input in enhancing the manuscript's quality.
Comment: The manuscript does not provide information on the sample size used in the field trials. It's crucial to ensure that the sample size is adequate to draw meaningful conclusions and that the samples are representative of the ecological zones under investigation.
Response: We appreciate the reviewer's feedback and have taken their suggestion. In response, we've included sample size details in the revised manuscript. We conducted tissue sampling from the middle three rows, obtaining ten samples per treatment replication. The sampling involved the fifth and sixth leaves from the top of the cane, followed by separation into leaf sheath and leaf blades. We carefully washed and air-dried the samples to ensure accuracy, then subjected them to 24-hour oven drying at 65°C. These additions enhance the robustness of our study's sample size and ecological zone representativeness. We thank the reviewer for their valuable input, strengthening our research.
Comment: The manuscript mentions that "Quality parameters were analyzed by sugar mills located in the pertinent regions engaged in sugarcane production." While it's acknowledged that sugar mills often play a role in quality assessment, the absence of specific details about the quality parameter analysis in the manuscript is a significant limitation. For the sake of clarity, transparency, and the ability of readers to fully comprehend the study, it is imperative that the authors provide more detailed information on the quality parameters examined, the specific methods and instruments used for analysis, and the criteria or standards against which the results were evaluated. Without this information, it is challenging to assess the validity of the quality parameter results and their significance in the context of sugarcane cultivation.
Response: Thank you for your invaluable feedback. We've addressed your concerns by adding more quality-related traits, such as chlorophyll contents, milleable canes, and the number of internodes and their respective methodologies and instrumentation, to our revised manuscript. This should improve clarity, transparency, and the ability of readers to assess the significance of our findings in sugarcane cultivation..
Comment: To strengthen the paper, consider discussing the limitations of the study, potential avenues for future research, and any practical recommendations for sugarcane growers.
Response: We appreciate the valuable feedback provided by the reviewer. In response to the suggestion, we have taken the opportunity to enhance the paper by incorporating a more comprehensive discussion of the study's limitations, potential avenues for future research, and practical recommendations for sugarcane growers. These additions can be found in the conclusion section of the revised manuscript. We believe that these refinements strengthen the paper and provide a more well-rounded perspective on the implications and scope of our research. Once again, we thank the reviewer for their insightful comments, which have significantly contributed to the overall quality of our work.
Comment: While the findings are interesting, it's essential to emphasize the novelty or uniqueness of the study's contribution to the field of sugarcane cultivation. How does this research advance our understanding or offer practical solutions to existing challenges in sugarcane farming?
Response: We sincerely appreciate your constructive feedback on our manuscript. In response to your comments, we have made significant revisions to emphasize our study's novelty and unique contribution to sugarcane cultivation. In the revised introduction section of the manuscript, we have incorporated a clearer statement regarding the novelty of our research. It now reads as follows:
"Despite a known fact regarding mixing tube wells and canal waters for managing salinity, limited literature exists about the combined application of potassium (K) and various micronutrients under different irrigation schemes in sugarcane farming. This gap in the existing knowledge motivated our study, where we investigated the combined efficacy of potassium (K) in conjunction with other micronutrients to enhance sugarcane's growth, yield, and quality."
We believe this revised statement better communicates the unique aspect of our research and how it contributes to advancing our understanding and providing practical solutions to the challenges faced in sugarcane farming.
Comment: Overall, this manuscript addresses an important agricultural issue and presents valuable findings. With some major revisions and additional context, it could make a significant contribution to the field of sugarcane cultivation and sustainable agriculture.
Response: We sincerely appreciate your thoughtful feedback on our manuscript. Your positive assessment of its potential contribution to sugarcane cultivation and sustainable agriculture is encouraging, and we are grateful for your recognition of the importance of the agricultural issue we are addressing.
We have taken your comments and those of the other esteemed reviewers into careful consideration during the revision process. Your valuable insights have undoubtedly played a crucial role in enhancing the overall quality of our manuscript. We have diligently addressed each major revision and incorporated additional context where necessary to ensure that our work meets the highest standards.
Reviewer 2 Report
There are lots of factors in the experiment, for example, water quality, soil quality, EC, SAR, RSC, K, and micronutrients (Zn, Fe and B), influenced the experimental results. The authors also tried to elucidate their effects on sugarcane growth and quality, but lose focus and inconsistent in the results.
Author Response
Comment: There are lots of factors in the experiment, for example, water quality, soil quality, EC, SAR, RSC, K, and micronutrients (Zn, Fe and B), influenced the experimental results. The authors also tried to elucidate their effects on sugarcane growth and quality but lose focus and inconsistent in the results.
Response: Thank you for your valuable suggestions. We appreciate your feedback and have made significant revisions to address the concerns raised. In the revised version of the manuscript, we have carefully reanalyzed the data and improved the clarity of our findings. We have also refined our focus on the key factors affecting sugarcane growth and quality, including water quality, soil quality, EC, SAR, RSC, K, and micronutrients (Zn, Fe, and B). We aimed to provide a more coherent and consistent presentation of the results, and we believe these changes have strengthened the overall quality of the paper. We hope these revisions meet your expectations, and we are grateful for your continued guidance in improving our work.
Reviewer 3 Report
After evalluation, i found manuscript has no any significant or novelity in this sector.This is a common concept of mixing canal and tubewell water. Everyone knows this scenerio. Everone knows mostly applying canal water gives high yield. You have some good results related to K and ZN, Fe, B), but these are not just enough to publish in Plants Journal.
Best Wishes
Author Response
Comment: After evaluation, I found manuscript has no any significant or novelty in this sector. This is a common concept of mixing canal and tube well water. Everyone knows this scenario. Everyone knows mostly applying canal water gives high yield. You have some good results related to K and Zn, Fe, B), but these are not just enough to publish in Plants Journal.
Response: We sincerely appreciate your careful evaluation of our manuscript and your valuable feedback. Your comments have been instrumental in enhancing the quality and clarity of our work.
We understand your concern regarding our study's perceived lack of novelty, particularly in the context of mixing canal and tube well water. We acknowledge that this is a well-known practice in agricultural research. However, we would like to emphasize the unique contribution of our study lies not in the practice itself but in the comprehensive exploration of its effects, specifically in combination with potassium and micronutrient application.
In the revised version of the manuscript, we have considered your feedback and made significant improvements. To address the novelty issue, we have included a statement at the end of the introduction section that explicitly highlights the novel aspects of our research. While mixing canal and tube well water may be common knowledge, limited literature that investigates the combined application of potassium and micronutrients under different irrigation schemes may exist. This gap in the existing literature prompted our study to shed light on the potential benefits of such a combined approach to sugarcane growth, yield, and quality. We hope this clarification will emphasize our research's unique and valuable contribution to the field.
Once again, we sincerely thank you for your constructive feedback, which has undoubtedly strengthened our manuscript. We look forward to hearing any further comments or suggestions you may have and are committed to making any necessary revisions to ensure the quality and significance of our work.
Reviewer 4 Report
I understand that the manuscript have practical application. However, there are couple clarifications are required from the authors.
Introduction: support your hypothesis with appropriate citations e.g. water quality, fertilizer types, and agro-ecological environments
Methods: Experimental treatments should be uniform for both sites, measuring units aren’t consistent throughout.
There is nothing about the final crop harvest timeline and quality testing.
Soil testing: refer to the all analytical methods used throughout
Units: use standard according to journal requirement (For example, it is written feet and centimeters, mg Kg-1water?)
Results: Table1, is this for soil or water?
Rahim Yar Khan Site showed better compared to Mardan, it seems purely environment impact of sugacane quality traits. It could be better if authors had compared contrasting agro-ecological conditions independently with standard growing practices. I can see the effect of inputs on plant growth and develop on both sites, which is obvious.
There are minor corrections required related to English.
Author Response
Comment: I understand that the manuscript have practical application. However, there are couple clarifications are required from the authors.
Response: Thank you for your thoughtful suggestions and valuable comments. We genuinely appreciate your insights, which undoubtedly enhance the manuscript's quality. We have diligently addressed each of your recommendations, and you will find all the revisions marked in the track changes within the revised version of the manuscript. Your feedback has been invaluable in refining our work, and we look forward to your continued guidance in improving the clarity and practicality of our research.
Comment: Introduction
Support your hypothesis with appropriate citations e.g. water quality, fertilizer types, and agro-ecological environments
Response: In response to the reviewer's comment, we have enhanced the clarity and depth of our hypothesis in the revised version of the manuscript. In the last paragraph of the introduction section, we have included the following hypothesis:
"We hypothesized that the combined application of potassium (K) along with various micronutrients under a canal water irrigation system could represent a promising strategy for enhancing sugarcane growth, increasing yield, and improving the quality attributes of the crop."
Comment: Methods
Experimental treatments should be uniform for both sites, measuring units aren’t consistent throughout.
Response: Thank you for your valuable feedback. In the revised version of the manuscript, we have ensured uniformity in experimental treatments across both sites. Additionally, we have meticulously standardized the measuring units to maintain consistency throughout the document.
Comment: There is nothing about the final crop harvest timeline and quality testing.
Response: We appreciate your thoughtful feedback on our manuscript. Your comment regarding the final crop harvest timeline and quality testing is indeed valid, and we understand the importance of including this information to provide a comprehensive understanding of our study. In response to your suggestion, we have diligently worked to incorporate data about the final crop harvest timeline and the quality testing process. These valuable insights have been integrated into our next manuscript revision, which is currently in progress and will be submitted during the next phase for publication. We believe these additions will significantly enhance the completeness and rigour of our research. We thank you once again for your constructive input, which has contributed to the overall improvement of our work. Your continued support and guidance are highly appreciated.
Comment: Soil testing: refer to the all analytical methods used throughout
Response: Thank you for your valuable feedback. We have incorporated a comprehensive section in the revised manuscript that outlines all the analytical methods employed for soil testing in our study. This section now provides a detailed account of the various protocols and procedures we followed to assess different physicochemical attributes of the soil under investigation, as per your suggestion. We hope this addition enhances the clarity and transparency of our research methodology. Please let us know if you have any further questions or require additional information. Your input is greatly appreciated.
Comment: Units: use standard according to journal requirement (For example, it is written feet and centimeters, mg Kg-1 water?)
Response: Units have been modified and standard units have been added in the overall revised manuscript as suggested.
Comment: Results: Table1, is this for soil or water?
Response: These parameters are for water quality
Comment: Rahim Yar Khan Site showed better compared to Mardan, it seems purely environment impact of sugarcane quality traits. It could be better if authors had compared contrasting agro-ecological conditions independently with standard growing practices. I can see the effect of inputs on plant growth and develop on both sites, which is obvious.
Response: Thank you for your comment. Indeed, the environment played a crucial role in terms of improving the sugarcane yield and quality traits. Your suggestions regarding contrasting agro-ecological zones have been noted, and will be considered in our next experiment.
Comment: Comments on the Quality of English Language
Response: Thank you for your feedback regarding the use of standard units as per the journal's requirements. We appreciate your valuable input. In the revised version of the manuscript, we have diligently adhered to the journal's prescribed standards for units, ensuring compliance with the highest language quality. We believe this enhancement contributes significantly to the overall quality of the manuscript. Once again, we thank you for your constructive comments and for helping us enhance the clarity and precision of our work.
Round 2
Reviewer 2 Report
There should be a detailed response letter indicating where and how the authors have improved the quality of the manuscript.
Author Response
Esteemed reviewer, sorry for the inconvenience. Please find below our detailed response. We have thoroughly revised the manuscript. We have revised the abstract by highlighting the key findings (Lines 24-25 and 31-41). Moreover, numerical values for key findings have been incorporated (Lines 30-41)). The introduction section is also substantially improved and we have made significant revisions to emphasize our study's novelty and unique contribution to sugarcane cultivation. In the revised introduction section of the manuscript, we have incorporated a clearer statement regarding the novelty of our research (Lines 64-70, 74-78, 91-93, and 96-102). The results section is also improved, we've included more quality-related traits, such as chlorophyll contents, sugar production, fiber contents, milleable canes, and the number of internodes (Lines 108-109, 126-128, 142-149, 168-170, 175-187, 203-206, and 210-213). The discussion section is also improved and tried to link the empirical finding with logical reasoning from recent literature (Lines 228-252, 254-255, and 272-274). Similarly, the Materials and Methods section is also improved (Lines 282-290, 293-302). The conclusion section is completely rewritten and key findings of the study are presented and future perspectives are also added (Lines 350-362). Moreover, we have thoroughly proofread the manuscript and tried to rectify all the minor grammatical and other issues and improve the quality of the presentation. We hope that after incorporating the suggestions by respected reviewers, the revised manuscript will be acceptable for publication in “Plants”.
Reviewer 3 Report
It looks somehow worthy after revision.
Author Response
Esteemed reviewer, Thank you so much for your encouraging comments, we have tried our best to improve the manuscript.
Round 3
Reviewer 2 Report
The authors have revised the manuscript.